# Experimental Study and Finite Element Analysis on the Flexural Behavior of Steel Fiber Reinforced Recycled Aggregate Concrete Beams

**DOI:** 10.3390/ma15228210

**Published:** 2022-11-18

**Authors:** Qiaoyan Guan, Mengyu Yang, Ke Shi, Tao Zhang

**Affiliations:** School of Civil Engineering and Architecture, Zhengzhou University of Aeronautics, Zhengzhou 450046, China

**Keywords:** steel fiber reinforced recycled aggregate concrete, flexural behavior, crack development, finite element analysis

## Abstract

This paper reports on the flexural behavior of nine steel fiber reinforced-recycled aggregate concrete (SFRAC) beams through combined experimental and finite element analysis. The test parameters in this study include the steel fiber volume fraction, recycled aggregate replacement ratio, and concrete strength. The failure modes, crack development, load-deflection curves, and flexural bearing capacity of SFRAC beams are investigated in detail. The test results indicated that cracks and concrete crushing are formed in the compression zone of all specimens. The flexural bearing capacity of SFRAC beams increases with the increase of steel fiber volume fraction and concrete strength and the decrease of recycled aggregate replacement ratio. In addition, the results are compared with those of the finite element analysis. Based on the uniaxial compressive constitutive model of SFRAC, a new model for calculating the flexural bearing capacity of SFRAC beams is proposed. The prediction and test results are compared to evaluate the accuracy of the developed formula. The studies may provide a considerable reference for designing this type of structure in engineering practice.

## 1. Introduction

Recycled Aggregate Concrete (RAC) is a new green building material that is made by partially or entirely substituting natural aggregates with recycled aggregates. Recycled aggregates are acquired by processing waste concrete with a high building waste content. The development of the construction industry has led to a large reduction of natural sand and gravel and the consumption of natural resources. The use of recycled concrete mitigates this situation [1]. RAC has been widely applied in civil construction, and these properties have been studied in depth worldwide. Developed nations have extensively studied RAC and developed design and building guidelines [2,3]. In the Netherlands, the use of recycled aggregate derived from concrete and bricks as a road base is an ordinary practice [4]. Arisha and Gabr [5] appraised the feasibility of using construction and demolition waste materials, especially blends of RCA with recycled crushed bricks as unbound granular material for road construction in Egypt. Nayana and Kavitha [6] evaluated that for the production of one ton of natural aggregates, 0.046 tons of CO_2_ is emitted as in comparison to 0.0024 tons of CO_2_ emitted in the production of one ton recycled aggregates. In comparison with natural aggregates, recycled aggregates reduce carbon emissions by 23–28%.

However, compared with the natural coarse aggregate, the recycled aggregate will produce internal damage during the crushing process. With more significant porosity and crushing index, the old cement base usually wraps its surface. Therefore, there is a big gap between the performance of RAC and ordinary concrete. Existing studies show that the compressive strength, tensile strength, and flexural bearing capacity of RAC are lower compared to that of plain concrete with natural aggregates. Moreover, as the recycled aggregate replacement ratio increases, the macro mechanical properties of RAC decrease accordingly [7,8,9]. Similarly, CaSucCIO [10] studied the strength and elastic modulus of RAC and proved that the strength and modulus of RAC are lower than ordinary concrete. In addition, Gonzalez-Corominas et al. [11] believe that the performance of recycled aggregate decreases with the reduction of the quality of recycled aggregate. They believe that only recycled aggregate with concrete strength above 60 MPa and recycled aggregate replacement ratio that cannot exceed 50% can be used to prepare high-performance concrete. At present, the main application of RAC is applied to the roadbed and pavement [12] and is less applied to the bearing structure. The scope of the application of recycled aggregate is limited [13]. It is very necessary to carry out research on RAC members so as to form the design method of RAC members, and finally expand the application range of RAC [14,15,16,17]. 

Although RAC itself has certain defects, existing studies have proved that the defects of recycled aggregate can be compensated to some extent by adding steel fibers. Adding steel fibers to recycled aggregates can improve the mechanical properties of RAC and the force and deformation properties of RAC structures [13,18,19,20]. Gao et al. [21] found that Young’s modulus and stress–strain curves of steel fiber reinforced recycled aggregate concrete (SFRAC) after the addition of steel fibers were similar to those of natural coarse aggregate concrete. Zhao et al. [19], by adding up to 1.5% steel fibers, the compressive, tensile and flexural behavior of SFRAC were obviously improved and more premium than natural aggregate concrete, which made it eligible for structural application. The results of Carneiro [22] showed that the addition of steel fibers and recycled aggregates increased the mechanical strength and improved the fracture process of the concrete compared to the reference concrete. The stress–strain characteristics of the RAC were influenced by the recycled aggregate, which exhibited more fragile characteristics than the reference aggregate. With the addition of steel fibers, the toughness of the RAC, as measured by the slope of the decreasing branch of the stress–strain curve, increased, and its compressive properties were similar to those of fiber-reinforced natural aggregate concrete [23,24,25]. Gao et al. [26] evaluated the effect of the RCA and steel fibers on the durability-related properties (carbonation resistance, freeze-thaw resistance, and chloride-ion penetration) of concrete made with different steel fiber volume fractions and recycled aggregate replacement ratios. They concluded that the durability of SFRAC was similar to that of plain concrete. An experimental study by Chen et al. [27,28] showed that concrete compressive strength and stiffness were significantly reduced after exposure to high temperatures. The addition of steel fibers retarded the sprouting of cracks in RAC, helped prevent spalling, and significantly improved the flexibility and cracking properties of RAC at high temperatures, which is beneficial for the application of RAC in building construction. Gao et al. [18] conducted an experimental and analytical study on the force performance of steel fiber concrete axially compressed columns. He concluded that steel fibers could improve the mechanical properties of concrete mixes and also control the protective layer spalling of concrete columns while improving the strength and flexibility of SFRAC axially compressed columns. Zong et al. [29] concluded that it is convenient and practical to incorporate steel fibers in RAC to form SFRAC. Compared with RAC, the mechanical properties of SFRAC meet the requirements of practical engineering and effectively improve the safety and service life of RAC structures, thus increasing the recycling rate of construction and demolition waste. Steel fibers in the concrete matrix play a key role in bridging and anchoring effects, which can prevent crack development and thus change the brittle failure mode of RAC.

At present, there are relatively more studies on the mechanical properties of SFRAC and relatively fewer studies on the properties of SFRAC members. From the existing research results at home and abroad, the flexural bearing capacity of SFRAC beams is improved, the stiffness is increased, the deflection becomes smaller, and the crack resistance is improved compared with that of RAC beams. The force performance and reliability can reach the level of ordinary concrete beams [30], and the overall cost is lower than that of ordinary concrete beams in the same condition. The savings of recycled aggregate are 2.5 times the increased cost of incorporating steel fibers [13]. However, there is a lack of systematic research on the flexural behavior of SFRAC beams at home and abroad, and the effects of steel fiber volume fraction and recycled aggregate replacement ratio on the flexural behavior of SFRAC beams have not been systematically discussed. The calculation of the flexural bearing capacity of SFRAC beams is only a simple fitting of experimental data, which lacks universality. These lacking studies restrict the application of SFRAC to the structural level. In this paper, the effects of recycled aggregate replacement ratio, steel fiber volume fraction, and concrete strength on the flexural behavior of SFRAC beams are investigated by means of SFRAC beams’ normal section flexural behavior tests. Based on the SFRAC uniaxial compressive constitutive model, considering the effects of steel fiber volume fraction and recycled aggregate replacement ratio, the model and equations for the calculation of the SFRAC beams’ flexural bearing capacity is directly derived from the cross-sectional static equilibrium condition using the constitutive relationship.

## 2. Experimental Programs

### 2.1. Material Properties

Ordinary Portland cement (P.O 42.5) in terms of Chinese Standard GB 175-2007 [31] was used in the study. City tap water was used as the mixing water in the experiment. The coarse aggregate types include natural coarse aggregate and recycled coarse aggregate, which are composed of a continuous gradation with a particle size of 5–20 mm. Natural coarse aggregate was crushed limestone, and recycled coarse aggregate was crushed waste concrete sourced from scientific research waste beam with water–cement ratio of 0.39. Fine aggregate was natural medium sand with good gradation and a fineness modulus of 2.75. The aggregates are as presented in Figure 1 and their properties are given in Table 1. Hook-ended steel fibers with an average length of 35 mm and a nominal diameter of 0.55 mm (tensile strength of 1345 N/mm^2^, equivalent diameter of 63) were used to reinforce concrete beams. Polycarboxylic acid water reducer with a water-reducing rate of 25% was used to adjust the workability of concrete mixtures. Steel bars with diameters of 8 and 16 mm were used as longitudinal steel bars. The stirrup uses a steel bar with a diameter of 6 mm and a spacing of 150 mm. The mechanical properties of the steel bars are presented in Table 2. Each result value is based on the average of three tested samples of the steel bars.

### 2.2. Beam Specimen Characteristics

A total of nine SFRAC beams were designed to investigate the flexural behavior with the steel fiber volume fraction (0, 1.0%, 2.0%), recycled aggregate replacement ratio (0, 30%, 50%, and 100%), and concrete strength (C60, C45, and C30) as test parameters. All beams were with the same dimensions of 150 × 300 × 3000 mm^3^ with a clear span of 2700 mm. Steel bar 1 was used for the stirrup and erection bar in the compression zone, and steel bar 2 was used for the tensile longitudinal steel bar at the bottom, as shown in Figure 2a. All specimens were mixed by a horizontal forced mixer for 5 min, then poured into the formwork, and put on a vibration table to vibrate for 30 s to ensure compaction. All specimens were demolded after 24 h and cured after 28 d. The tests were performed after 28-day age.

The mixture proportions of SFRAC are summarized in Table 3. Due to the adhesion of old cement mortar on the surface of recycled coarse aggregate, compared with natural aggregate, the characteristics of small apparent density and large porosity and water absorption of recycled coarse aggregate should be considered in the mix design. According to the design method of ordinary concrete mixture proportion and considering the influence of water absorption of recycled aggregate, the mixture proportion of SFRAC in the test was designed [32]. The recycled aggregate is pre-wetting prior to concrete mixing. The pre-wetting quantity depends on the test value of recycled aggregate water absorption. 

### 2.3. Experiment Setup and Test Method

All specimens were tested for flexural behavior according to Chinese Standard GB/T50152-2012 [33]. The test equipment adopts a universal testing machine. In order to eliminate the influence of shear stress on the flexural behavior of the normal section of the specimen, the beams were tested under a simply supported four-point loading condition (Figure 2). Thus, pure-bending sections of beams with a length of 900 mm were formed between the two loading points of the distribution beam. All specimens were loaded using force control. The specific loading steps were as follows: The load was monotonically increased until the beam failed, during which the loading was sustained at every 2.5 kN increment for a while to observe cracking evolution. After cracking, the loading rate was increased to 5 kN per stage until the beam failed. During the loading, the strains of reinforcing bars and top concrete were collected. The YWC-100 type resistance displacement sensor and resistance strain gauge were used to measure the deflection of the beam and the concrete strain of the beam section, respectively. The loading method and measurement arrangement are shown in Figure 2. 

**Figure 2 materials-15-08210-f002:**
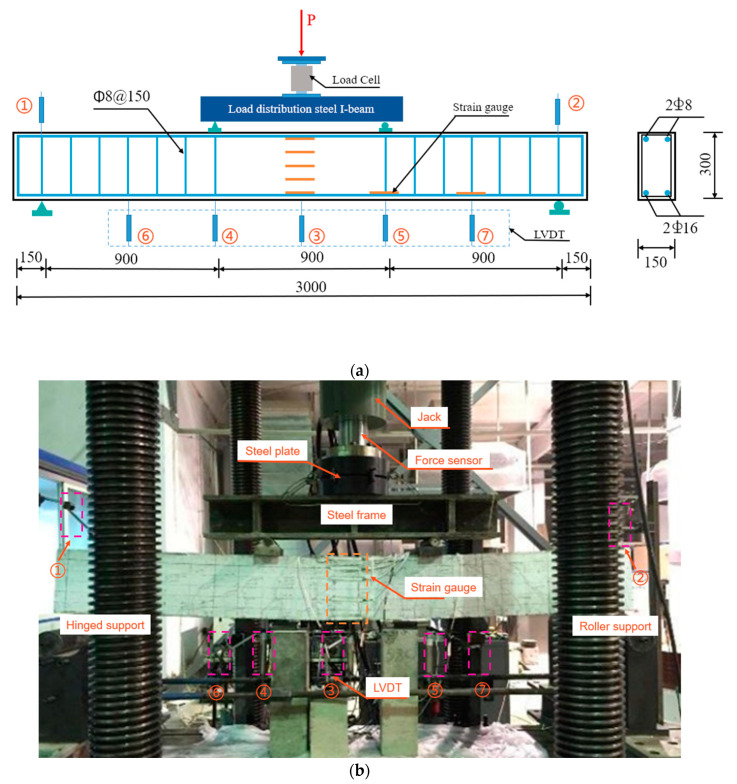
(**a**) Schematic diagram (Dimensions in mm), (**b**) test setup. (Note: The numbers from 1 to 7 in the Figure are displacement sensor).

## 3. Finite Element Analysis (FEA)

### 3.1. Finite Element Modeling

To investigate the effect of the steel fiber volume fraction, concrete strength, and recycled aggregate replacement ratio factors on the flexural behavior of SFRAC beams, the flexural behavior tests on eight SFRAC beams were simulated (see Figure 3). The model was divided into five components: Concrete, steel plate, hoop, compressive steel bars, and tensile steel bars. Concrete was modeled through solid three-dimensional (3D) elements (C3D8), whereas 2-node truss elements (T3D2) were employed for steel bars. The bond between the two materials was assumed perfect at their common nodal points. Different mesh sizes were searched to select the most suitable size (sensitivity analysis). The model was simulated with 25, 30, and 40 mm mesh sizes to check the convergence of the solutions (see Figure 4). Depending on the sensitivity analysis, a suitable mesh was adopted. The FEA model of 30 mm mesh size was used for all specimens, as given in Figure 3, keeping in mind that the selected mesh size should balance between accuracy in evaluated results and computational time. The boundaries are set according to the same restrictions as the actual test. Embedded regions can be used to integrate truss element parts into concrete solid element parts. Steel bar cage parts are selected for the embedded part, and concrete beam parts are selected for the main area. The coupling constraints are set at the bottom of the steel plate. The left support was constrained for the displacement degrees of freedom in the X, Y, and Z directions, while the right support was constrained for the displacement degrees of freedom in the Y and Z directions. The displacement control loading method was employed for loading.

### 3.2. Material Modeling

The concrete damage plastic (CDP) model available in ABAQUS was used to define the plastic properties of concrete. In addition to isotropic tensile and compressive plasticity, isotropic damage elasticity is also used to characterize the overall performance of concrete. The constitutive model of concrete under uniaxial compression stress–strain based on the relationship of SFRAC concrete in Gao et al. [34] is represented by Equation (1). The tensile stress–strain curve of recycled concrete of Lu et al. [35] was selected as the tensile constitutive model of concrete is represented by Equation (2). The CDP model also requires five additional parameters to define the failure criterion, namely the shape factor (k_c_), the stress ratio (f_b0_/f_c0_), the eccentricity (*ε*), the viscosity (μ), and the dilation angle (ψ). According to the recommendation of ABAQUS (2014), these five parameters are 0.667, 1.16, 0.1, 0.005, and 30, respectively. The sensitivity of viscosity parameters was analyzed in the finite element analysis and the most appropriate μ value was selected to balance the accuracy and efficiency of the calculation. The value of the viscosity parameter is taken as 0.005 based on numerous simulations performed to investigate the effect of this parameter on the results. For the main steel bar, an ideal elastic-plastic bilinear model was employed.
(1)$$\begin{array}{c} \left\{\begin{array}{l} y=\alpha_{1}x+(3-2\alpha_{1})x^{2}+(\alpha_{1}-2)x^{3},x\leq 1\\ y=\frac{x}{\alpha_{2}{\left(x-1\right)}^{2}+x},x>1 \end{array}\right.\\ \alpha_{1}=(0.9237-0.03473\delta_{R})(1.824-0.09056\lambda_{f})\\ \alpha_{2}=(1.281-0.8786\delta_{R})(1.672-0.8086\lambda_{f}) \end{array}$$
(2)$$\begin{array}{c} \sigma =\left(1-d_{t}\right)E_{t}\varepsilon \\ d_{t}=\left\{\begin{array}{l} 1-\rho_{t}\left(a_{1}+\left(1.5-1.25a_{1}\right)x+\left(0.25a_{1}-0.5\right)x^{5}\right),x\leq 1\\ 1-\frac{\rho_{t}}{\alpha_{t}{\left(x-1\right)}^{1.7}+x},x>1 \end{array}\right.\\ \alpha_{1}=1.2(1+0.0065r-0.0057\lambda_{sf})\\ \alpha_{t}=\frac{0.312f_{t}^{2}}{1-0.0042r+0.0515\lambda_{sf}}\\ \lambda_{sf}=v\left(L/D\right) \end{array}$$

### 3.3. Validation of the FEA Model

Figure 5 gives the comparisons of the load vs. deflection curves between experimental and corresponding FE results, which can reflect the deformation process, including the elastic stage, the working stage with cracks, and the damage stage. At first, it can be found that there are limited differences in flexural stiffness between experimental and FE results at the initial loading stage, and the flexural stiffness of the FE results is slightly greater than that of experimental results. This is because the crack of concrete led to a reduction in the stiffness of the specimens in experiments, whereas the concrete crack under low stress is neglected in FE model. Besides, the predicted curves agree reasonably well with the experimental results, and the discrepancies of the flexural load between experimental and FE results are less than 15%. In addition, it can be seen from Figure 6 that the failure mode captured by the FEA model is consistent with that found in the actual test. Figure 6 demonstrates that the failure mode of FE analysis resembles that of experimental results. 

In summary, the FE modeling approach, material constitutive model and boundary condition, as mentioned above, are proved to be reasonable and can be used to further investigate the flexural performance of SFRAC beams.

## 4. Experimental Results and Discussions

### 4.1. Crack Development and Failure Mode

Figure 7 shows the crack development in all beams. No significant cracks were observed in the early stages of loading. The cracks first become visible at the bottom of the beams at loads between 12 and 25 kN. It should be observed that cracks emanated from the tension zone of the beams for all specimens, and their nature was hair-like. The cracks staggered upwards from the bottom, concentrating mostly toward the middle of the span of the beams, and were all flexural, as indicated in Figure 7. More cracks developed and spread within the span of the beams with the increase in loading. Microcracks propagated gradually towards the loading points with the increase in loading. 

The concrete in the compression zone (top) of the beams was crushed until the beam failed. Generally, the neutral axis depth kept reducing following crack development upon load application, so that a corresponding reduction of the compression zone. The steel bars yielded first, followed by a localized crushing of the concrete in the compression zone, as indicated in Figure 6. This type of failure is usually considered as a ductile mode of failure. Although the nature of crack development in NAC and RAC beams is similar, SFRAC beams have fewer and lesser severe cracks than all the other beams. Evidently, the presence of steel fibers led to a more ductile mode of failure. The test results are shown in Table 4. 

**Figure 6 materials-15-08210-f006:**
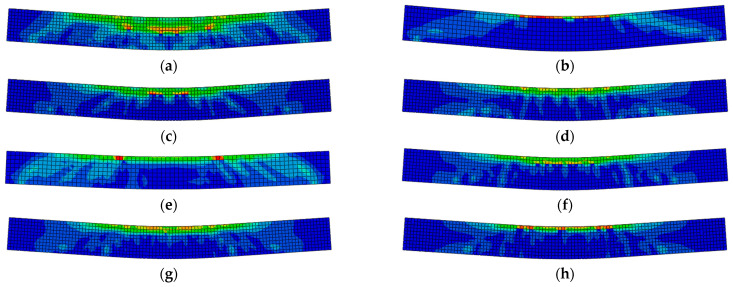
The damage diagram of SFRAC beams. (**a**) C30R50F1, (**b**) C45R0F1, (**c**) C45R50F0, (**d**) C45R50F2, (**e**) C45R30F1, (**f**) C45R50F1, (**g**) C45R100F1, (**h**) C60R50F1.

**Figure 7 materials-15-08210-f007:**
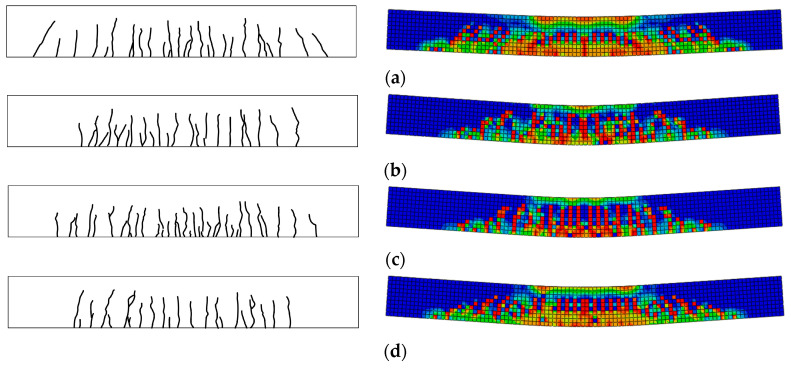
Crack development of test and FE results. (**a**) C30R50F1, (**b**) C45R0F1, (**c**) C45R50F1, (**d**) C45R50F0, (**e**) C45R50F2, (**f**) C45R30F1, (**g**) C45R100F1, (**h**) C60R50F1.

### 4.2. Load-Deflection Curves

The variation law of the load-midspan deflection is basically the same for each beam. It can be seen from Figure 8 that there are two obvious turning points in the load-midspan deflection curve so that the mechanical and deformation process of the beam specimen can be divided into three stages, the elastic stage, the working stage with cracks, and the damage stage. In the first phase, the concrete in the tensile zone has not yet cracked. The load increases linearly with the increase of deflection. The bending moment and tensile strain of the beam section are small, and the tensile stress and initial stiffness of the tensile zone are shared by the RAC, steel fibers, and longitudinal steel bar. The steel fiber volume fraction, concrete strength, and recycled aggregate replacement ratio have almost no effect on the deformation performance of the beams in this stage. In this stage, the stiffnesses of all beams are close to each other. In the second phase: The phase from after cracking until the longitudinal steel bar yields. The load-deflection curves of SFRAC beams were similar to those of ordinary concrete beams, but the slope of the curve did not change significantly after cracking due to the inhibition of the development of concrete cracks by the incorporation of steel fibers. With the increase of steel fiber volume fraction, the development of cracks was inhibited, so that the ductility of SFRAC beams was well improved. In the third stage: Damage stage. When the damage state was reached, the longitudinal steel bar yields, and the load-deflection was horizontally related. From the yielding of the tensile steel bar to the final crushing of the concrete, the increase in the load was slight, and the increase in the beam deflection was significant, which indicates the good ductility of the beam. Figure 9a shows that the deflection decreases with the increase of steel fiber volume fraction at the same load level. However, the steel fibers in the concrete tend to aggregate when the steel fiber volume fraction is larger than 1.0%. Figure 9b indicates that the stiffness of SFRAC beams tends to increase with the increase of steel fiber volume fraction in the elastic phase and cracking phase, and the ductility of SFRAC beams was better after the yielding of steel bar. Figure 9c illustrates that stiffness tends to decrease with the increase of the recycled aggregate replacement ratio in the elastic phase. The stiffness in the cracking phase was basically unchanged, and the ductility of SFRAC beams was lower than that of ordinary steel fiber concrete beams with the same concrete strength after the steel bar yields. 

### 4.3. Flexural Bearing Capacity

As can be seen from Figure 10, the normal cross-sectional flexural bearing capacity of the RAC beams was basically the same as that of the ordinary concrete beams with the same concrete strength. The normal cross-sectional flexural bearing capacity of the SFRAC beams was greater than that of the ordinary concrete beam and the RAC beams with the same concrete strength. The steel fiber across the crack can bear part of the tensile force. After the concrete cracks in the tensile zone, the flexural bearing capacity of SFRAC beams is larger than that of ordinary concrete and recycled concrete beams. The higher the concrete strength, the greater the bond between the recycled concrete and steel fibers in the tensile zone, and thus the greater the flexural bearing capacity.

The addition of steel fiber can prevent crack development, increase strength, and increase toughness. When steel fiber volume fraction is 1.0% and 2.0%, the flexural bearing capacity of the SFRAC beams is 19.5% and 24.8% higher than that of RAC beams, respectively. It shows that under the same conditions, the flexural bearing capacity of SFRAC beams increased with the increase of steel fiber volume fraction. Moreover, the flexural bearing capacity of the initial crack increased with the increase of the ultimate compressive strain of concrete in the compression zone. The study also found that under the same conditions, the deflection of test beams decreased with the increase of steel fiber volume fraction. The reason is that steel fibers in the tensile zone after cracking can continue to bear part of stress, thereby preventing crack development and increasing the cracked section’s stiffness. To sum up, it can be seen that the addition of steel fibers increased the ultimate compressive strain of RAC, increased the toughness of RAC, and effectively reduced the deformation of the test beams and the development of cracks.

The flexural bearing capacity of the normal section of SFRAC beams decreased with the recycled aggregate replacement ratio increased. The reduction of flexural bearing capacity of SFRAC beams is all smaller than that of steel fiber concrete beams with the same concrete strength and larger than that of plain concrete beams with the same concrete strength, as shown in Figure 10c. Under the same conditions, the flexural bearing capacity of SFRAC beams was all reduced in different magnitudes compared to steel fiber concrete beams. The flexural bearing capacity of SFRAC beams was reduced by 27.6%, 25.1%, and 31.0% when the recycled aggregate replacement ratio was 30%, 50%, and 100%, respectively. The steel fibers in the tensile zone provided tensile force when the beams reached the ultimate bending state. The magnitude of tensile force provided by the steel fibers was closely related to the bond strength of the steel fibers and concrete. Bond strength and tensile strength provided by steel fibers decreased with increasing recycled aggregate replacement ratio. At the same time, the compressive concrete strength in the beam compression area was also reduced so that the flexural bearing capacity of SFRAC beams is lower than that of ordinary steel fiber concrete beams.

## 5. Analytical Models for Flexural Bearing Capacity

### 5.1. Fundamental Assumptions 

#### 5.1.1. Flat Section Assumptions

The flat section assumption is generally applicable to continuous homogeneous elastic members. SFRAC beams are characterized by non-homogeneous material and the presence of cracks, and strictly speaking, the flat section assumption does not apply at localized locations of damage. The measured mean strain values for the midspan section are shown in Figure 11. It is clear that the midspan section of the SFRAC beams meets the assumption of flat section deformation.

#### 5.1.2. Contribution of SFRAC in the Tension Zone

Ordinary concrete beams generally do not consider the role of concrete in the tensile zone. The steel fibers in SFRAC make it possible for tensile stresses to remain in the specimen after cracking, which effect on the member’s breaking moment cannot be ignored. The steel fibers in the SFRAC provide tensile stress, which influences the ultimate moment of the specimen and cannot be ignored.

#### 5.1.3. Material Stress–Strain Relationship

The stress–strain relationship of reinforced concrete structural materials is shown in Figure 12. The stress–strain relationship for SFRAC is based on the constitutive model of SFRAC under uniaxial compression in Gao et al. [34]:(3)$$\left\{\begin{array}{l} y=\alpha_{1}x+(3-2\alpha_{1})x^{2}+(\alpha_{1}-2)x^{3},x\leq 1\\ y=\frac{x}{\alpha_{2}{\left(x-1\right)}^{2}+x},x>1 \end{array}\right.$$
(4)$$\alpha_{1}=(0.9237-0.03473\delta_{R})(1.824-0.09056\lambda_{f})$$
(5)$$\alpha_{2}=(1.281-0.8786\delta_{R})(1.672-0.8086\lambda_{f})$$

Stress–strain relationships for steel bar:(6)$$\sigma_{s}=E_{s}A_{s}\leq f_{y}$$

### 5.2. Calculation of Flexural Bearing Capacity

In the existing literature, the equivalent rectangular stress graph method calculates the normal section flexural bearing capacity of SFRAC members. This method needs to determine two characteristic parameters. They are consistent, but the compressive stress and strain curves of concrete adopted by the national codes are the same. The method in this paper directly adopts the stress and strain curve instead of the equivalent rectangular stress graph method, which avoids the problem of determining the characteristic parameters by experiments. It provides a new calculation method for the design of the normal section of the beam. Stress and strain distributions of the strengthened beams under flexural failure is as shown in the Figure 13.

The compressive stress–strain curve of SFRAC is expressed as:(7)$$\left\{\begin{array}{l} \sigma_{fr}=f_{c}\left[\alpha_{1}\frac{\varepsilon }{\varepsilon_{0}}+(3-2\alpha_{1}){\left(\frac{\varepsilon }{\varepsilon_{0}}\right)}^{2}+(\alpha_{1}-2){\left(\frac{\varepsilon }{\varepsilon_{0}}\right)}^{3}\right],\varepsilon \leq \varepsilon_{0}\\ \sigma_{fr}=f_{c},\varepsilon >\varepsilon_{0} \end{array}\right.$$
where
(8)$$\alpha_{1}=(0.9237-0.03473\delta_{R})(1.824-0.09056\lambda_{f})$$

Simplify Equation (7) as:(9)$$\begin{array}{c} \sigma_{fr}=E_{1}\varepsilon +E_{2}\varepsilon^{2}+E_{3}\varepsilon^{3}\\ E_{1}=\frac{\alpha_{1}f_{c}}{\varepsilon_{0}}\;{}_{\left(1\right)}\\ E_{2}=\frac{\left(3-2\alpha_{1}\right)f_{c}}{\varepsilon_{0}^{2}}\\ E_{3}=\frac{\left(\alpha_{1}-2\right)f_{c}}{\varepsilon_{0}^{3}} \end{array}$$

Assuming that the height of the concrete in the compression zone is *x*, it is known from the theory of elasticity that the strain in the compression zone is:(10)$$\varepsilon =\frac{\varepsilon_{cu}}{x}y$$
where $\varepsilon_{cu}$ is the ultimate compressive strain of SFRAC. It can be seen from the test that the ultimate compressive strain of steel fiber is basically the same as that of ordinary concrete after adding steel fiber to RAC. Take the value according to 6.2.1 in GB50010-2010 [36]:(11)$$\varepsilon_{cu}=0.0033-(f_{cu,k}-50)\times 10^{-5}\leq 0.0033$$
where $f_{cu,k}$ is the standard value of the compressive concrete strength cubes.

From the section static equilibrium conditions, it follows that:(12)$$\int_{0}^{x}\sigma bdy=f_{y}A_{s}+\sigma_{f}(h-x)$$
(13)$$M_{u}=\int_{0}^{x}\sigma bdy(h_{0}-\frac{x}{2})-\sigma_{f}\left(h-x\right)\left(\frac{h-x}{2}-a_{s}\right)$$
where $f_{y}$ is the steel bar stress in the tension zone, $a_{s}$ is the distance from the resultant point of the steel bar in the compression zone to the edge of the compression zone, $\sigma_{f}$ is the equivalent tensile stress of the SFRAC in the tension zone. From the analysis of the test data, it can be seen that the tensile stress of the SFRAC in the tension zone is related to the recycled aggregate replacement ratio and the characteristic admixture of the steel fiber. By fitting the test data in this paper, we can get:(14)$$\sigma_{f}=\left(1-0.803\delta_{R}\right)\left(1+4.775\lambda_{f}\right)f_{t}$$
where is the design value of the tensile strength of ordinary concrete with the same strength grade as SFRAC.

Substituting Equations (9) and (10) into Equations (12) and (13):(15)$$\frac{bx}{\varepsilon_{cu}}\left[E_{1}\frac{x^{2}}{2}+E_{2}\frac{x^{3}}{3}+E_{3}\frac{x^{4}}{4}\right]=f_{y}A_{s}+\sigma_{f}(h-x)$$
(16)$$\frac{bx}{\varepsilon_{cu}}\left[E_{1}\frac{x^{2}}{2}+E_{2}\frac{x^{3}}{3}+E_{3}\frac{x^{4}}{4}\right](h_{0}-\frac{x}{2})-\sigma_{f}(h-x)(\frac{h-x}{2}-a_{s})=M_{u}$$

The formula for the flexural behavior capacity of a normal section of a flexural beam when damage to a reinforced beam is reached:(17)$$\left[\frac{bE_{1}\varepsilon_{cu}}{2}+\frac{bE_{2}\varepsilon^{2}{}_{cu}}{3}+\frac{bE_{3}\varepsilon^{3}{}_{cu}}{4}\right]x=f_{y}A_{s}+\sigma_{f}(h-x)$$
(18)$$\left[\frac{bE_{1}\varepsilon_{cu}}{2}+\frac{bE_{2}\varepsilon^{2}{}_{cu}}{3}+\frac{bE_{3}\varepsilon^{3}{}_{cu}}{4}\right]x(h_{0}-\frac{x}{2})-\sigma_{f}(h-x)(\frac{h-x}{2}-a_{s})=M_{u}$$

### 5.3. Formula Validation

The ultimate flexural bearing capacity of concrete beams reinforced with steel fibers depends to a large extent on their mode of destruction. ACI 544.4R-88 [37] provides a calculation method regarding the flexural bearing capacity of reinforced beams based on the mode of destruction of concrete beams reinforced with steel fibers. For beams with crushed concrete damage, the nominal flexural bearing capacity can be calculated by the following equation:(19)$$M_{n}=\left[A_{s}f_{y}(d-\frac{a}{2})+\sigma_{t}b(h-e)(\frac{h}{2}+\frac{e}{2}-\frac{a}{2})\right]$$
(20)$$a=\frac{\sigma_{t}b(h-e)+A_{s}f_{y}}{0.85f_{c}^{\prime }b}$$
(21)$$e=\left[\varepsilon_{f}+0.003\right]c/0.003$$
(22)$$\sigma_{t}=0.00772l/d_{f}\rho_{f}F_{be}$$

Calculation formula based on the Chinese code for the structural design of steel fiber concrete(JTG/T 465-2019) [38]:(23)$$M_{f}=f_{fc}bx(h_{0}^{}-\frac{x}{2})+f_{y}^{\prime }A_{s}^{\prime }(h_{0}-a_{s}^{\prime })-f_{ftu}bx_{t}(\frac{x_{t}}{2}-a)$$

According to the calculation formula of SFRAC normal section flexural bearing capacity proposed in this paper and ACI 544.4R-88 [37] and the calculation formula of flexural bearing capacity suggested by the Chinese code (JTG/T 465-2019) [38], the test and a total of eight concrete beam specimens with the same loading mode, plastic failure mode, and more detailed data than in this paper are calculated. The test results are compared with the calculation results. The mean value, mean squared error, and coefficient of variation of the ratio between the calculated value and the measured value of the normal beam section flexural bearing capacity using the formula proposed in this paper are 1.06, 0.065, and 0.061, respectively. The mean value, mean squared error, and coefficient of variation of the ratio between the calculated value and the measured value of the test by ACI 544.4R-88 [37] are 1.08, 0.085, and 0.079. The mean value, mean squared error, and coefficient of variation of the ratio between the calculated value and the measured value of the test by the JTG/T 465-2019 [38] method are 0.91, 0.051, and 0.048. Comparisons of the flexural bearing capacity of SFRAC as illustrated in the Table 5 and Figure 14. The results show that the calculation method proposed in this paper has good validity and accuracy. The calculation formula proposed in this paper can effectively predict the flexural bearing capacity of SFRAC beams.

## 6. Conclusions and Discussion

This paper aims to investigate and evaluate the flexural behavior of SFRAC beams through combined experimental, numerical, and theoretical studies in terms of the failure modes, crack development, load-deflection curves, and flexural bearing capacity. The parametric analysis is also conducted. Based on the limited results from the current study, the following conclusions can be drawn:The stressing process of SFRAC beams is similar to that of ordinary concrete beams, which undergoes the elastic stage, the working stage with cracks, and the damage stage. However, the slope of the curve cannot change significantly after cracking due to the inhibition of the development of concrete cracks by the incorporation of steel fibers. With the increase of steel fiber volume fraction, crack development is inhibited so that the ductility of SFRAC beams is well improved.The flexural bearing capacity of SFRAC beams increases with the increase of steel fiber volume fraction and concrete strength, while the decrease of recycled aggregate replacement ratio. The flexural bearing capacity of the SFRAC beams is greater than that of the normal concrete beams and recycled concrete beam with the same concrete strength. FE results are established and verified by the experimental results. Reasonable agreement is achieved between experimental and corresponding FE results. It demonstrated that the FEA model can fairly predict the flexural behavior of the SFRAC beams, including the failure modes and crack development.A model for calculating the flexural bearing capacity of SFRAC beams is established with consideration of the influence of steel fiber volume fraction and recycled aggregate replacement ratio. What is more, this calculated model directly adopts the stress and strain curve instead of the equivalent rectangular stress graph method, which avoids the problem of determining the characteristic parameters by experiments. The predicted results exhibited good agreement with experimental results. Due to the limitation of time and the author’s level, this paper only studied the flexural behavior of SFRAC beams under static load, and further research is needed to study the flexural behavior and establish the analytical model of SFRAC beams under fatigue load.

## Figures and Tables

**Figure 1 materials-15-08210-f001:**
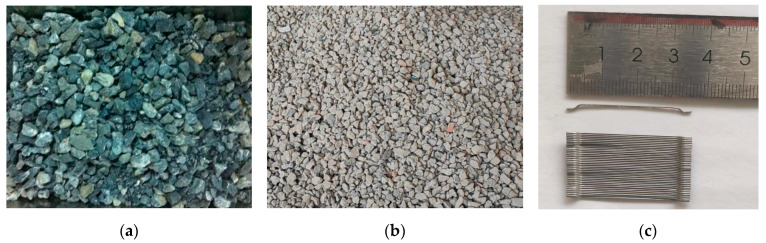
The appearance of materials: (**a**) Natural coarse aggregate, (**b**) recycled coarse aggregate, (**c**) steel fibers.

**Figure 3 materials-15-08210-f003:**
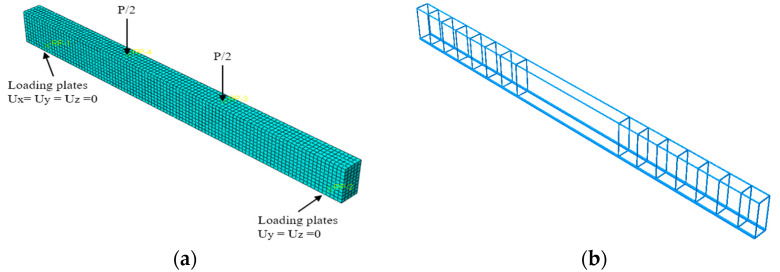
The FEA model. (**a**) Concrete and boundary consideration, (**b**) rebar cage.

**Figure 4 materials-15-08210-f004:**
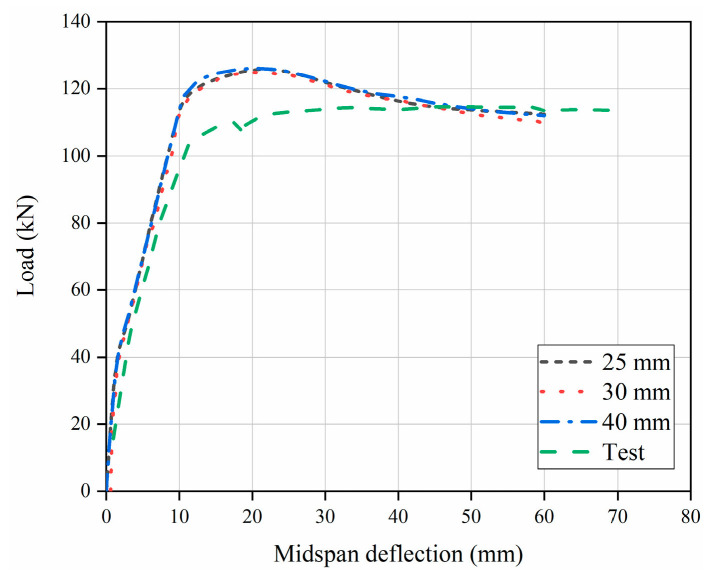
Mesh size effect of specimen C30R50F1.

**Figure 5 materials-15-08210-f005:**
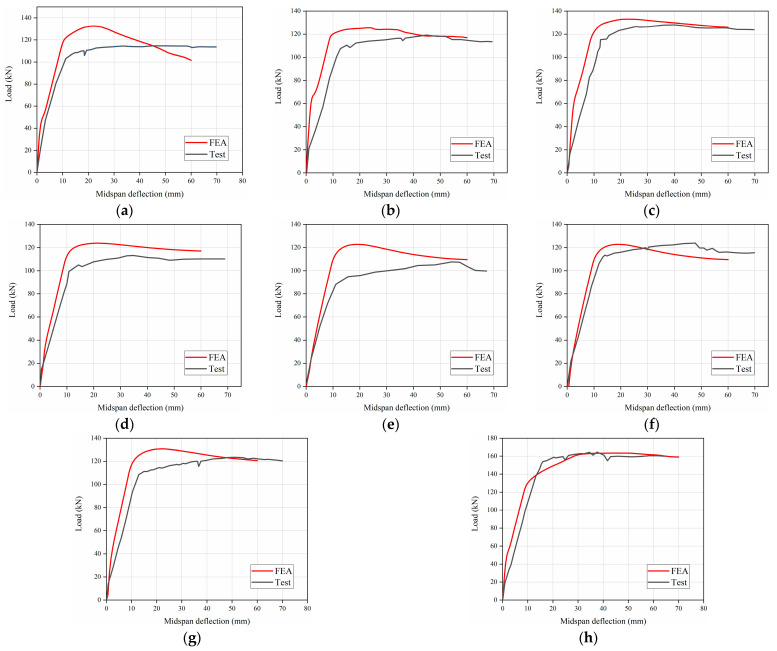
Load–deflection curves between test and FE results. (**a**) C30R50F1, (**b**) C45R30F1, (**c**) C45R50F2, (**d**) C45R100F1, (**e**) C45R50F0, (**f**) C45R50F1, (**g**) C60R50F1, (**h**) C45R0F1.

**Figure 8 materials-15-08210-f008:**
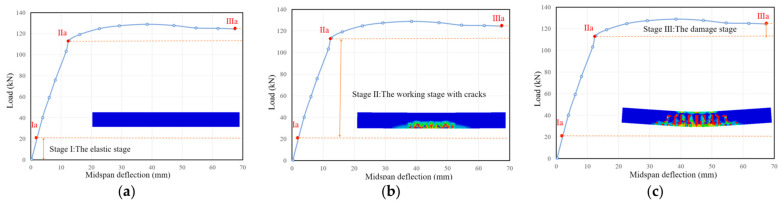
Three stages of beam force and deformation. (**a**) The elastic stage, (**b**) the working stage with cracks, (**c**) the damage stage.

**Figure 9 materials-15-08210-f009:**
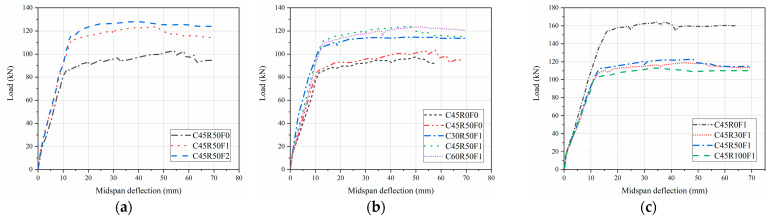
Load–deflection curves of the tested beams. (**a**) Steel fiber volume fraction, (**b**) concrete strength, (**c**) recycled aggregate replacement ratio.

**Figure 10 materials-15-08210-f010:**
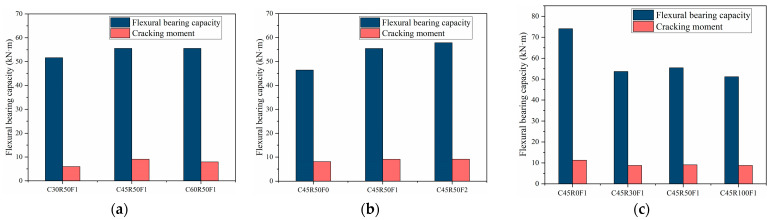
Flexural bearing capacity of SFRAC beams. (**a**) Concrete strength, (**b**) steel fiber volume fraction, (**c**) recycled aggregate replacement ratio.

**Figure 11 materials-15-08210-f011:**
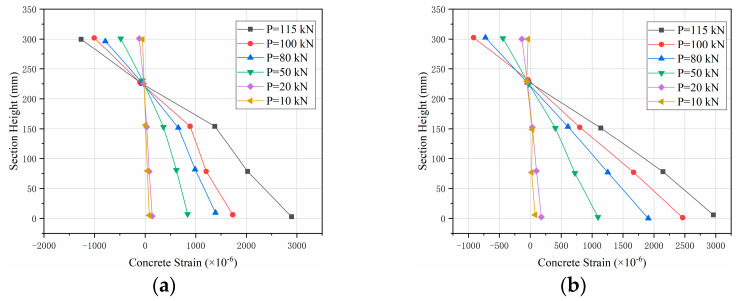
Strain distribution of midspan section along section height. (**a**) C45R50F1, (**b**) C45R50F2.

**Figure 12 materials-15-08210-f012:**
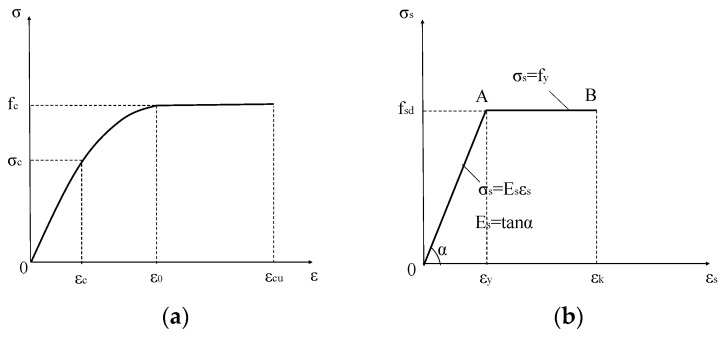
Stress–strain relationship of reinforced concrete structural materials. (**a**) Compression of concrete, (**b**) rebar in tension.

**Figure 13 materials-15-08210-f013:**
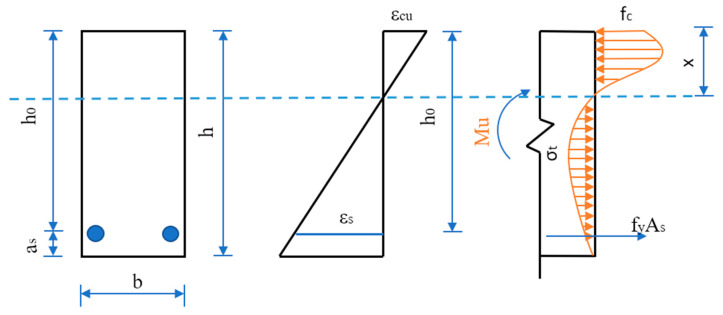
Stress and strain distributions of the strengthened beams under flexural failure.

**Figure 14 materials-15-08210-f014:**
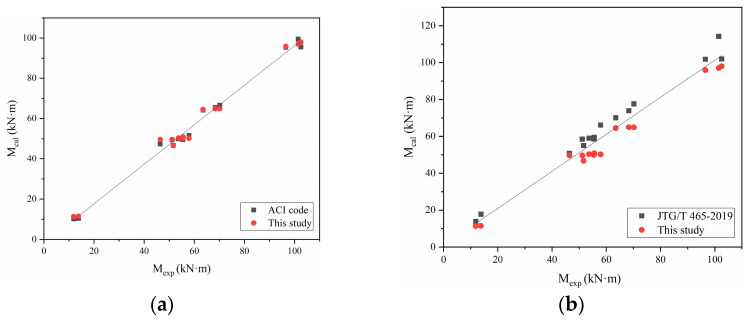
Comparisons between experimental results (*M*_exp_) and predicted values (*M*_cal_). (**a**) *M*_exp_ versus *M*_cal_ of this study and ACI code, (**b**) *M*_exp_ versus *M*_cal_ of this study and JTG/T 465-2019.

**Table 1 materials-15-08210-t001:** Physical properties of aggregate.

Aggregate Type	Apparent Density (kg/m^3^)	Bulk Density (kg/m^3^)	Water Absorption (%)	Crush Index (%)	Porosity (%)
RCA	2660	1410	3.73	13.5	47
NCA	2730	1360	0.6	12	40
NFA	2640	1560	--	--	40

**Table 2 materials-15-08210-t002:** Mechanical properties of steel bar.

Category	*d*_s_ (mm)	*f*_y_ (MPa)	*f*_st_ (MPa)	*δ*_st_ (%)
Steel bar 1	8	325	493	41.3
Steel bar 2	16	483	623	24.2

Note: *d*_s_ is diameter; *f*_y_ is yield strength; *f*_st_ is tensile strength; *δ*_st_ is elongation at fracture.

**Table 3 materials-15-08210-t003:** Mixture proportions of SFRAC.

Number	*r*	*v* _sf_	Concrete Mixing Proportion (kg/m^3^)						
			Water	Cement	NCA	RCA	Sand	Water Reducer	Additional Water
C30R50F1	50%	1%	166	302	540	540	884	3.02	19.06
C45R0F0	0%	0%	166	415	1024	0	839	4.15	0
C45R0F1	0%	1%	166	415	1024	0	839	4.15	0
C45R30F1	30%	1%	166	415	717	307	839	4.15	10.85
C45R50F0	50%	0%	166	415	512	512	839	4.15	36.16
C45R50F1	50%	1%	166	415	512	512	839	4.15	18.08
C45R50F2	50%	2%	166	415	512	512	838	4.15	18.08
C45R100F1	100%	1%	166	415	0	1024	839	4.15	36.16
C60R50F1	500%	1%	166	553	479	479	783	5.53	16.89

Note: *r* is recycled aggregate replacement ratio; *v*_sf_ is steel fiber volume fraction.

**Table 4 materials-15-08210-t004:** Beam cracking load, ultimate load, and basic mechanical properties of concrete.

Specimen Label	*f*_ts_ (MPa)	*f*_c_ (MPa)	*E*_c_ (×10^4^ MPa)	*f*_cr_ (kN)	*f*_mu_ (kN)
C30R50F1	3.73	27.7	3.16	13.37	114.81
C45R0F0	2.11	38.8	3.35	18.23	97.62
C45R0F1	4.54	40.5	3.64	25	164.7
C45R30F1	4.89	39.4	3.45	19.5	119.28
C45R50F0	1.69	31.7	3.13	18.01	103.15
C45R50F1	4.09	39	3.3	20.17	123.31
C45R50F2	6.1	40.5	3.55	20.28	128.73
C45R100F1	3.98	34.4	2.97	19.61	113.7
C60R50F1	3.58	38.9	3.6	17.68	123.54

Note: *f*_ts_ is splitting and tensile strength; *f*_c_ is; *E*_c_ is elasticity modulus of SFRAC; *f*_cr_ is cracking load; and *f*_mu_ is ultimate load.

**Table 5 materials-15-08210-t005:** Comparisons of the flexural bearing capacity of SFRAC.

Data Sources	Specimens	*b* (mm)	*h* (mm)	*r*	*v* _sf_	*M*_exp_ (kN·m)	*M*_cal_ (kN·m)	*M*_JTG_ (kN·m)	*M*_ACI_ (kN·m)	*M*_exp_/*M*_cal_	*M*_exp_/*M*_JTG_	*M*_exp_/*M*_ACI_
This study	C30R50F1	150	300	0.5	0.01	51.66	46.69	55.00	46.68	1.11	0.94	1.11
	C45R30F1	150	300	0.3	0.01	53.68	50.39	59.03	49.88	1.07	0.91	1.08
	C45R50F1	150	300	0.5	0.01	55.49	49.98	58.60	49.54	1.11	0.95	1.12
	C45R100F1	150	300	1	0.01	51.17	49.66	58.46	49.43	1.03	0.88	1.04
	C45R50F0	150	300	0.5	0	46.42	49.64	50.78	47.41	0.94	0.91	0.98
	C45R50F2	150	300	0.5	0.02	57.93	50.27	66.13	51.69	1.15	0.88	1.12
	C60R50F1	150	300	0.5	0.01	55.59	50.90	59.49	50.24	1.09	0.93	1.11
Cheng et al. [39]	FB-1	200	300	0.6	0.01	63.5	64.51	70.06	64.16	0.98	0.91	0.99
	FB-2	200	300	0.6	0.01	68.33	64.90	73.89	65.51	1.05	0.92	1.04
	FB-3	200	300	0.6	0.02	70.21	64.93	77.58	66.63	1.08	0.90	1.05
Liu et al. [40]	L-1	200	400	0	0	102.53	98.01	101.93	95.54	1.05	1.01	1.07
	L-2	200	400	1	0	96.53	96.53	101.82	95.42	1.01	0.95	1.01
	L-3	200	400	1	0.01	101.48	101.48	114.12	99.43	1.05	0.89	1.02
Emmanuel E. et al. [41]	NAC	80	180	0.6	0	11.9	11.36	13.9	10.17	1.05	0.86	1.17
	SFRRAC	80	180	0.6	0.005	13.8	11.50	17.77	10.43	1.20	0.78	1.32
AVG										1.06	0.91	1.08
COV										0.06	0.048	0.079

Note: *M_e_*_xp_ is experiment value of flexural bearing capacity, *M*_cal_ is calculated value of flexural bearing capacity, AVG is average value, and COV is coefficient of variation.

## Data Availability

Data are available on request from the authors.

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
