# Peer review of "Experimental Study and Finite Element Analysis on the Flexural Behavior of Steel Fiber Reinforced Recycled Aggregate Concrete Beams"

_materials, 2022, doi:10.3390/ma15228210_

Round 1
Reviewer 1 Report
The submitted manuscript has a typical structure, where the topic addressed is very interesting.
The manuscript have also of potential interest to journal readers.
The research task combines experimental research in combination with numerical modeling. This approach is very positive. The use of recycled aggregates in value-added solutions is also an interesting research area.
However, the manuscript must be improved before publication so that the quality of the information presented is of the required quality.
The introduction section must be fundamentally improved. The information provided is insufficient. All information about the current status of the solved problem for all solved parts should be given. Specifically, it is about the bearing capacity of reinforced concrete beams, the use of recycled aggregate (handling, mechanical properties, use) and information about fiber reinforced concrete, as well as numerical modeling of beams using the finite element method. Interesting articles include https://doi.org/10.1016/j.proeng.2015.06.222 (Finite Element Modeling and Identification of the Material Properties of Fiber Concrete); https://doi.org/10.3390/ma15207204 (Effect of Moisture Condition of Brick–Concrete Recycled Coarse Aggregate on the Properties of Concrete) Authors must clearly state the motivation and originality of the topic addressed.
Provide more detailed information about the standards for the classification of cement, concrete and concrete reinforcement. How many concrete reinforcement tests were performed? Add a static mathematical description? Add photo. What is the surface reinforcement?
Improve the description and add a high-quality photo instead of Fig. 3. Fig.3 - photo of very poor quality. What the supports look like and where the load is area. Add photos and description.
Numerical modeling is a very important part of the research problem. The results from the performed are interesting, but it is necessary to add more information about the calculation.
It is not entirely clear from the manuscript what kind of fiber reinforced concrete tests were done and the procedure for identifying the input parameters for the calculation. Please provide more information. Ideally, list the input information in tables.
Fig. 5 - The description of the graph axes is difficult to read. Improve
Fig. 7 - Scale - the scale is hard to read, improve it
Fig. 9 - The description of the graph axes is difficult to read. Improve
Fig. 11 - The description of the graph axes is difficult to read. Improve
The discussion of the results in the context of the current state, limitations of the solution and new knowledge must also be improved.
The manuscript and the research topic are very interesting. The manuscript will be very interesting after editing.
Author Response
请参阅附件

Reviewer 2 Report
Please have a look on the attached file, In my opinion extensive work is required.

Reviewer 3 Report
I congratulate all the authors for the research, for the scientific content of the work. Authors need to address the following queries to improve the quality of the article further.
1. The abstract is a summary of the introduction, materials and method, results and conclusion. This order needs to be followed. The methodology, results (quantifying data) and conclusion component of the abstract should be properly captured.
2. “Gao et al. found that Young's modulus” mention reference number.
3. Add quantitative results in the literature reviews.
4. “Gao [14] conducted an experimental and analytical study” Reference 14 has multiple authors. So, correct the same.
5. Highlight contribution of the study to knowledge gap/specific problem. Clearly.
6. “fineness modulus of 2.75.” What is the unit?
7. “Table 2. Mechanical properties of reinforcing steel.” Correct the unit from “Mpa” to “MPa”.
8. “Figure 2. Details of longitudinal reinforcement and cross-section of the beams.” Mention All dimensions are in mm.
9. “ Figure 3 (a)” Mention All dimensions are in mm.
10. Improve the quality of figure 5.
11. “located in the middle third of the beams span” What do you mean by middle third?
12. “Overall, the SFRAC beams had the highest average number of cracks, but the crack strength was much lower than that of the other beams. It was also observed that while the cracks produced by the SFRAC beams were mostly disarticulated under additional loading, the RAC beams were usually continuous.” Add the proper and clear reason for these statements.
13. “It can be seen through Figure 9 that” is it figure 8 or 9? Rectify the same.
14. “under uniaxial compression in Gao [26]” Reference 26 has multiple authors. So, correct the same.
15. Kindly reconcile the conclusion with the study objectives.
16. What are the practical implications of this study and the future directions? kindly state
Round 2
Reviewer 1 Report
The modifications made have a partial character.
The overall level needs to be improved of the manuscript.
Recommendation:
1) Include a discussion section in the manuscript.
2) Improve the conclusions in the context of the current state of the issue being addressed
3) In the current state, there is not enough information on the chosen numerical modeling methods and materials - it is necessary to improve
4) The information on the part of numerical modelling is not sufficiently described and cannot be evaluated sufficiently in this regard
Reviewer 2 Report
Please see the attached file

Reviewer 3 Report
Authors have addressed all the queries. Article may be accepted in the present form.
Author Response
Dear Reviewer:
Thank you very much for your recognition. Your suggestion has improved our article.
Round 3
Reviewer 1 Report
The manuscript has been improved.
The authors responded well to comments.
The article will potentially be of interest to readers.
Reviewer 2 Report
Thanks for taking my comment positively